# Using Decision Science for Monitoring Threatened Western Snowy Plovers to Inform Recovery

**DOI:** 10.3390/ani11020569

**Published:** 2021-02-22

**Authors:** Bruce G. Marcot, James E. Lyons, Daniel C. Elbert, Laura Todd

**Affiliations:** 1Pacific Northwest Research Station, U.S. Forest Service, Portland, OR 97205, USA; 2Patuxent Wildlife Research Center, U.S. Geological Survey, Laurel, MD 20708, USA; 3Tennessee Ecological Services Field Office, United States Department of Interior, South Atlantic-Gulf Interior Region, Fish and Wildlife Service, 446 Neal Street, Cookeville, TN 38501, USA; daniel_elbert@fws.gov; 4Newport Field Office Newport, Pacific Region, Department of Interior, U.S. Fish & Wildlife Service, Newport, OR 97365, USA; lauraltodd@gmail.com

**Keywords:** western snowy plover, population monitoring, decision science, strategy, structured decision making

## Abstract

**Simple Summary:**

We developed a decision-analysis evaluation of a suite of nine alternative strategies for monitoring federally Threatened populations of Western Snowy Plovers (*Charadrius nivosus nivosus*) along the Pacific Coast, US. The species is increasing in numbers as a result of successful recovery plan implementation efforts, and is no longer feasible to conduct absolute censuses of birds and nests, as well as track productivity, fate, and predation events at every nest. What is needed is a statistically sound and economically feasible sampling approach to continue monitoring plover populations and informing management decisions that advance recovery for the species. We convened an eight-person technical team of plover monitoring experts to score the nine alternative strategies on a set of six categories of monitoring objectives such as maximizing the accuracy of determining the adult population size. Scoring consisted of ordinal scales of performance measures related to the recovery criteria for the species, and to other criteria related to monitoring reporting. We calculated overall scores among the team members, and explored how different objective weights influenced which monitoring strategies were best. Several monitoring strategies stood out as having the highest utility, depending on the importance given to cost, which we subsequently conveyed to the US Fish and Wildlife Service, responsible for monitoring as well as for consideration when choosing a standard monitoring sampling strategy throughout all the plover recovery units.

**Abstract:**

Western Snowy Plovers (*Charadrius nivosus nivosus*) are federally listed under the US Endangered Species Act as Threatened. They occur along the US Pacific coastline and are threatened by habitat loss and destruction and excessive levels of predation and human disturbance. Populations have been monitored since the 1970s for distribution, reproduction, and survival. Since the species was federally listed in 1993 and a recovery plan was approved under the US Fish and Wildlife Service in 2007, recovery actions have resulted in growing populations with increased presence at breeding and wintering sites throughout their Pacific Coast range. This success has created logistical challenges related to monitoring a recovering species and a need for identifying and instituting the best monitoring approach given recovery goals, budgets, and the likelihood of monitoring success. We devised and implemented a structured decision analysis to evaluate nine alternative monitoring strategies. The analysis included inviting plover biologists involved in monitoring to score each strategy according to a suite of performance measures. Using multi-attribute utility theory, we combined scores across the performance measures for each monitoring strategy, and applied weighted utility values to show the implications of tradeoffs and find optimal decisions. We evaluated four scenarios for weighting the monitoring objectives and how risk attitude affects optimal decisions. This resulted in identifying six strategies that best meet recovery needs and were Pareto optimal for cost-effective monitoring. Results were presented to the US Fish and Wildlife Service, responsible for monitoring as well as for consideration to ensure consistent monitoring methods across the species’ range. Our use of structured decision-making can be applied to cases of other species once imperiled but now on the road to recovery.

## 1. Introduction

Western Snowy Plovers (*Charadrius nivosus nivosus*, WSPs) occur along the Pacific coastline of North America, and are year-round residents throughout most of this range. Populations of this subspecies of snowy plover are particularly vulnerable to habitat loss and destruction, which can exacerbate levels of predation and human disturbance, especially during the breeding season. In 1993, the US Fish and Wildlife Service (USFWS) listed this subspecies as Threatened [1], and delineated nearly 160 breeding, wintering, and migration areas important for the recovery of the species within six Recovery Units along the coasts of Washington, Oregon, and California (Figure 1). A recovery plan was approved in 2007 [2] with the goal of ensuring that long-term conservation is provided for the species by meeting criteria for minimum thresholds of population size and sustainable nest productivity. Specific recovery objectives included (1) increasing population numbers, (2) conducting intensive, ongoing, and sustainable management for the species and its habitat, and (3) monitoring populations and threats to determine the success of recovery actions and refine management activities as needed.

The recovery plan for the WSP Pacific coast population [2] incorporated three specific criteria that, when met, would signal that the species would be ready for consideration of removal from the USFWS list of threatened species (also see Appendix A for further details):

Criterion 1. Monitoring shows that an average of 3000 breeding adults distributed among six recovery units have been maintained for a minimum of 10 years.

Criterion 2. A yearly average productivity of at least one fledged chick per male has been maintained in each recovery unit in the last five years prior to delisting.

Criterion 3. Mechanisms have been developed and are in place to assure long-term protection and management of breeding, wintering, and migration areas to maintain the subpopulation sizes and average productivity specified in Criteria 1 and 2.

Any successful strategy for monitoring Pacific coast populations of WSPs will provide data pursuant to these criteria to inform when the species has achieved recovery and no longer warrants being listed as Threatened under the Endangered Species Act. Determining the best successful strategy to standardize monitoring WSPs would entail a novel approach to evaluating alternative monitoring approaches.

WSP populations have been monitored since the 1970s for distribution, reproductive effort, and survival. Monitoring is based on the recovery criteria to determine adult population size, productivity, and whether management measures maintain stable or increasing populations. Since the recovery plan was instituted, monitoring has been based on annual full census counts of numbers of breeding adults, nests, and young, and has revealed growing populations with increased presence at sites across their Pacific Coast range. In Oregon, for example, monitoring data have shown significant population increases from a low of 28 adults in 1992, to more than 500 adults in recent years [3]. Essentially, the populations are increasing in response to the consistent implementation of effective recovery actions, and therefore all adults, young, nests, eggs, and their fates, can no longer be fully censused and counted due to personnel and budgetary constraints, necessitating a statistically rigorous sampling approach.

Such recovery successes to date have created logistical challenges to monitoring increasing sizes and numbers of populations with limited budgets, and a need to adapt monitoring approaches that maintain the consistency of data collection and the ability to support scientifically defensible decisions based on those data. Our analysis addressed the question of what cost-effective approach could be used that would provide the necessary monitoring of recovery criteria and best inform management decisions. We used a structured decision-making approach to determine which monitoring approaches might suffice or excel to meet recovery objectives with consideration for cost [4,5]. Decision analysis is used widely in areas of natural resource management and conservation [6], including monitoring of rare or at-risk species [7,8]. Studies of WSPs have provided much useful information on chick survival [9], adult survivorship and population trend [10], efficacy of predator control on WSP nest success [11,12,13], and impacts of invasive species management on WSP nesting [14]. Much of this information can further help inform the need for, and structure of, an effective sampling approach to monitoring WSPs along the Pacific Coast that could also help reduce monitoring costs.

The purpose of our work was to use expert knowledge from WSP managers and field monitoring crews to conduct a formal decision analysis of alternative monitoring sampling strategies by which to inform USFWS. Specifically, our objectives were to: Determine alternatives for monitoring WSPs and for measuring recovery criteria with efficient sampling approaches whenever possible; apply a novel, rigorous decision science approach to evaluate monitoring alternatives; identify cost-effective monitoring strategies and advise USFWS on the degree to which each alternative addresses recovery objectives; and promote best monitoring approaches to consider for instituting and standardizing across the recovery regions.

## 2. Materials and Methods

### 2.1. Framing the Decision Problem and Identifying Objectives

We followed the PrOACT approach for structured decision making [4,15], which involves a sequence of five steps: Framing the decision problem, articulating objectives, identifying action alternatives, predicting the consequences of those alternatives in terms of the stated objectives, and finally, evaluating tradeoffs among competing objectives. Specifically, our analysis of consequences and tradeoffs relied on the multi-attribute utility theory to measure the relative desirability of potential monitoring strategies [16,17].

We framed the decision problem as a choice of the most effective and time- and cost-efficient monitoring strategy providing data necessary for conservation and management. The regulatory authority for monitoring WSPs exists in the Endangered Species Act. The USFWS ultimately determines the listing status of WSP, including any changes to the status as supported by the best available science. Therefore, the USFWS is the decision maker who will choose a monitoring strategy for data necessary to evaluate the listing status. Agency decisions for public natural resource management affect many stakeholders. A government agency may ultimately make the decision, but stakeholder groups can significantly influence this choice by their actions and potential actions [17]. WSP monitoring is accomplished by a large consortium of public (federal, Tribal, state, and local agencies) and private stakeholders. Thus, we convened a WSP Technical Team (hereafter, “team”) consisting of eight WSP monitoring and recovery experts with experience in a wide range of monitoring methods. In addition, organizing and conducting risk assessment sessions with a panel of agency managers and decision-managers was beyond the scope of the present project. The team experts were drawn from each state within the species’ current range (Figure 1) to ensure that a range of options for the sampling strategies was considered, and because they brought together the diversity of expertise most appropriate to understanding the various logistical and budgetary challenges associated with monitoring increasing WSP populations. As the organizers and facilitators of this effort, we brought our own long expertise to methods of modeling, monitoring, decision science, and expert elicitation and paneling [5,13,18,19,20,21,22], including means of identifying and avoiding sources of panelist and facilitator bias [20,23].

Including a range of experience and perspectives helps at every PROACT step. We followed a structured approach to expert panel knowledge elicitation [5,24] to help ensure equal and fair contributions by each team member, and to elicit their expert knowledge individually rather than asking the group to reach consensus [19]. Retaining individual responses has the advantage of identifying team members who might have a unique experience, or who work in a geographic location or ecological setting different than those of others’, and whose input may otherwise be discounted as an outlier rather than a given value as important knowledge to consider.

To identify objectives for the monitoring strategy decision, we first reviewed with the team the specific recovery objectives and criteria set forth in the WSP recovery plan [2]. The aim of monitoring populations of WSPs is to provide measures of the above recovery plan criteria, potentially gather additional data on nest fate, and to inform management of the species. Monitoring based on the recovery criteria would determine the size of the adult population, productivity, and whether management measures are serving to maintain stable or increasing populations. Next, through extended dialogue with the team, we agreed on a set of six monitoring objectives for the decision at hand (Figure 2). Two of the objectives were related to measuring two recovery criteria (population size and fledgling productivity). Objectives related to monitoring other demographic parameters (annual survival and nest fate) and monitoring to inform management decision making were also included (Figure 2). Finally, given the motivation to find cost-effective approaches that would include the data necessary for conservation and management, cost was also included as an objective. Two of the objectives (understanding nest fate and monitoring to inform management decisions) included subobjectives (Figure 2, Table 1).

For each objective and subobjective, we worked with the team members to identify performance measures by which the alternative monitoring strategies could be rated for each objective (Table 1, Appendix A). Performance measures are used to clarify the meaning of, and quantify achievement toward, objectives in the decision context [25]. Our performance measures were constructed (ordinal) scales that included levels of accuracy, effectiveness, and information transfer. Accuracy of estimated adult population size, fledgling productivity, and survival of adults and juveniles was gauged as a composite of various levels of bias and precision, with higher accuracy denoted with lower bias and higher precision of expected field monitoring outcomes under each monitoring strategy. Effectiveness of determining nest fate was gauged by comparison of known nest failures and predator identification with historic determination levels. Timeliness of information transfer to managers and decision-makers was gauged by frequency with which observations are relayed, and availability of information was gauged by whether the information is conveyed by written reports or verbally (Table 1, Appendix A).

### 2.2. Alternative Monitoring Strategies and Consequences

Through a structured query format, we next solicited ideas to develop a set of potential monitoring strategies for the species that would each, to varying degrees, provide information pertinent to the species monitoring objectives. We specifically encouraged creative thinking in devising alternative monitoring strategies. This process involved first identifying a variety of different ways to accomplish each of the six monitoring objectives (Figure 2), and then using a strategy table [4] to select individual activities and organize them into a monitoring strategy that addressed all six objectives (Table 2). We initially identified 10 strategies but eliminated one as being too similar to others, for a final set of nine monitoring strategies to further consider (see Appendix A for details of the monitoring strategies).

To understand the consequences of adopting each of the different potential monitoring strategies, we elicited judgments from the team using a modified Delphi method [26], a systematic process to elicit judgment from a group of experts with an iterative, facilitated discussion in which the participants provide judgments and then review, discuss, and revise their answers as desired. We had each team member, using a spreadsheet template (Appendix A), first independently score each strategy as to how well it would meet each monitoring objective’s performance measure. The scoring consisted of five-class ordinal scoring scales [9,26,27,28,29] for monitoring objectives 1-5.1, and a two-class ordinal scale for monitoring objective 5.2 (Appendix A). The ordinal levels on the scale were defined as, and pertained to, how well the performance measure of each monitoring objective could be met. We also had each team member score the cost of each monitoring strategy using a relative scale whereby they specified 100 points to what they would each identify as the highest-cost strategy, and then values of 100 or less for the percentage of the costs of each other strategy relative to the highest-cost strategy (Appendix A). We initially explored more detailed ways to gauge costs, as discussed further below, but we encountered complications among WSP monitoring teams and locations in leveling how monitoring costs are accounted. Thus, we settled on a simpler, relative scale for expressing the expected cost of each sampling strategy.

Additionally, we had each team member record their rationale for why they scored each strategy as they did. This took the form of their noting key strengths and evidence, key uncertainties, and key assumptions for their scoring of each monitoring sampling strategy and each monitoring objective, including cost (Appendix A).

After scoring the monitoring strategies by objectives and by costs, we summarized and shared results with the team members, and engaged them in a structured disclosure, discussion, and opportunity for having them individually present their scores and rationale, and to ask questions of one another. We kept all summaries and presentations of the team’s scores and rationale individually anonymous, referring to team members by randomly assigned letter codes. This procedure allowed each team member to individually decide what they wished to disclose, and also served to encourage them to freely express their judgments and rationale without worrying about being held individually accountable for them. This procedure was important for avoiding several potential types of biases present in expert panel procedures, such as bandwagoning (agreeing with the majority regardless of independent thinking), domineering (one panelist dominating discussions by dint of personality or intimidation), and herding (a facilitator guiding the group to one idea and minimizing others) [23]. Then, we conducted a second round of scoring, again independently, in which the panel members were given the opportunity to either retain or amend their scores depending on what they may have heard and learned from others during the disclosure and discussion session. The second round of scoring constituted the final set of scores that we brought into the next phase of analysis.

### 2.3. Evaluating Tradeoffs Using Objective Weights and Risk Attitudes

In the final step of the PrOACT sequence [15], we evaluated tradeoffs using weights for the objectives and utility functions representing risk attitudes. A utility function transforms an outcome stated in terms of the performance measure to a measure of utility between 0 and 1. One utility function is assessed for each performance measure. The worst outcome possible on the performance measure has a utility equal to 0 and the best outcome on the performance measure has a utility equal to 1 (see Figure 3 for example utility functions). Furthermore, the shape of a utility function reflects a decision maker’s risk attitude [16,17]. A straight-line utility function indicates a risk-neutral decision maker, in which each incremental increase in the performance measure corresponds to a proportional increase in utility (Figure 3). Concave and convex curves, which are often implemented as exponential functions, are used for decision makers that are risk averse and risk tolerant, respectively [27]. The shape of the function, concave or convex, reflects a decision maker’s “risk premium”, the amount of value the decision maker perceives from avoiding a less favorable or bad outcome [28]. For a risk neutral decision maker, the risk premium is zero. For risk averse and risk tolerant decision makers, the risk premium is positive and negative, respectively. We implemented utility functions in Excel using the Visual Basic for Applications (VBA) code for exponential functions [27]. Lyons et al. [29] provided the equation and the parameters of our exponential functions (Appendix A).

The multi-attribute utility theory relies on a linear additive model [16,30] to combine the utility from multiple objectives into one measure of relative desirability for each alternative in the decision context. The linear additive model can be written as:(1)$$U\left(x\right)=\sum_{i=1}^{n}w_{i}u\left(x_{i}\right)$$
where $w_{i}$ are the weights that sum to one and represent the relative importance of performance measure *i*; $u\left(x_{i}\right)$ is the single-dimension utility of outcome $x_{i}$ on performance measure *i*, and $U\left(x\right)$ is the summed weighted (total) utility for the alternative.

We used two approaches to help decision makers and stakeholders better understand tradeoffs: (1) A Pareto efficiency analysis, and (2) an evaluation of four scenarios for various combinations of weights and risk attitudes. A Pareto efficiency analysis is a graphical technique for understanding tradeoffs among two objectives [31]. The result of a Pareto analysis is not one solution but a set of non-dominated alternatives. A non-dominated alternative is one in which it is not possible to improve performance for one objective without giving up performance on another objective. We conducted a Pareto analysis with cost vs. summed utility provided by objectives 1–5 (based on straight-line utility functions, i.e., risk neutral).

Our second approach to evaluate tradeoffs used two scenarios for objective weights and two scenarios for risk attitudes (risk neutral and risk averse). Analyzing the implications of different weights applied to the monitoring objectives provides an end-user, such as a manager or decision-maker, with the capacity to determine how the monitoring sampling strategies might perform differently if some monitoring objectives were emphasized over others. The first scenario for how a decision maker might assign weights to objectives used equal weight on the six fundamental objectives. For objectives with subobjectives, the weight was apportioned equally among subobjectives (Table 1). The second scenario for weights used more weight (80%) on the objectives related to monitoring the recovery criteria (objectives 1 and 2) and cost (objective 6). We evaluated this scenario, which we call “more on mandates,” given the imperative to monitor the recovery criteria and the importance in this decision context of logistical and budgetary constraints (Table 1). We also incorporated risk attitudes by finding solutions that would be optimal for two types of decision makers: A risk neutral decision maker compared to a risk averse decision maker. A formal elicitation of risk attitude with decision makers was beyond the scope of this study. We used an informal assessment of risk with the team and chose a moderately risk-averse utility function for two objectives (accuracy of population size and cost). We evaluated changes in the preferred monitoring strategy with two levels of risk management (Figure 3, Appendix A). The combination of two scenarios for weights and two for risk attitudes resulted in four different scenarios to evaluate.

### 2.4. Analysis of Team Scores

We compiled the team’s second-round scores into a summary of ranges and means of score values of monitoring objectives and costs among the team members. The smallest ranges of values (maximum score value − minimum score value) among the team members signaled which combinations of monitoring objectives and monitoring sampling strategies were scored most similarly among the team members, whereas the largest range values signaled the greatest disparity in scoring. During the structured disclosure and discussion session following the first round scoring, we prompted the team members to discuss their rationale for scoring particularly pertaining to the monitoring objectives and strategies with the greatest range of score values, so that all members could learn from each other’s experience and reasoning.

### 2.5. Sensitivity Analysis of Sampling Strategies

In applications of the multi-attribute utility theory, weights in the linear additive model can be thought of as the substitution rate among objectives [16]. Weights are integral to effective tradeoffs in decisions with multiple objectives. In any decision context, tradeoffs are personal to the decision maker and there are no universal rules that determine the appropriate weights. Therefore, they are entirely subjective in nature [32,33]. Given the importance of weights when there are multiple objectives, it is helpful to evaluate the sensitivity of any decision analysis results to the weights assigned by the decision maker. We conducted a sensitivity analysis to understand how changes in the weight assigned to cost would affect the rank order of the monitoring strategies. We chose cost due to the uncertainty of the cost estimates and because budgetary and logistical constraints have emerged as a concern as WSP populations have increased over time. We varied the weight assigned to cost from 0 to 1 in increments of 0.2 and determined the rank order of the strategies at each increment. When varying the weight assigned to cost, the weights for other objectives were adjusted to remain in their original proportions yet still sum to one. This analysis measures the robustness of the preferred strategy to changes in weight assigned to cost.

### 2.6. Summary and Presentation of Final Results

We presented the final results to the team, and ultimately to USFWS managers of the WSP recovery process, of how the nine sampling strategies scored relative to one another for meeting each monitoring objective and total utility for all the objectives, and relative cost, under various weighting and risk attitude scenarios. USFWS decision-makers can review the strategies with the overall highest utility and the ability of each strategy to meet recovery objectives in light of budget constraints and desired weights for the objectives.

## 3. Results

### 3.1. Monitoring Sampling Strategies

Extended discussions with the team resulted in identifying nine potential alternative monitoring sampling strategies (Table 2). The strategies were designed to span a wide range of methods, including sampling intensities, levels of accuracy of determining specific population conditions and parameters, and expected levels of cost and personnel engagement. The strategies would, to varying degrees, meet the needs of the specific monitoring objectives including the recovery criteria (details are presented in Appendix A). The strategies differed by methods used to estimate the adult population status, fledgling productivity, survival of individuals, determination of nest fate, and communication of findings to managers (Table 2).

### 3.2. Scoring of Monitoring Strategies by Objectives

The initial inspection of the mean and normalized raw scores (Figure 4A,B) revealed that the sampling strategies varied widely in how well they would perform in meeting the monitoring objectives. The lowest-cost sampling strategies—(D) Minimal I Marked Population, and (E) Minimal II Effort/Resources—performed the poorest, whereas several higher-cost strategies—particularly (B) Varied Population Sizes, and (G) Marked Population—performed much better.

The team members differed in their scoring in a number of instances, as denoted by the range of their individual scores (Figure 4C). This was most notable with their spread of score values for the first four monitoring objectives for four of the sampling strategies. During the structured disclosure and discussion sessions, we used this information to review the definitions of these specific monitoring objectives and their performance measures, along with the descriptions of the sampling strategies. This provided all the team members with the same understanding for their second round of scoring, so as to avoid lexicon (linguistic) uncertainty, which is another potential source of bias in expert paneling [23]. Thereafter, any further apparent disparities in scoring among the team members were appropriate and authentic outcomes of members differing in their field experience, in how their institutions conduct monitoring, and other practical matters. To this end, we retained all such differences in scoring in our analyses, and we did not call for consensus or eliminate outlier score values.

### 3.3. Pareto Efficiency and Multi-Attribute Utility

The Pareto efficiency analysis identified six non-dominated alternatives (Figure 5). The non-dominated alternatives included low cost options such as (D) and (E) —the minimal effort strategies—which were approximately half as expensive as the most expensive strategy ((G) Marked Population). Strategies (B) Varied Population Sizes and (C) Variable Plover Densities resulted in nearly identical estimated costs and total utility and were among the non-dominated solutions. Strategies (F) Marked Individuals, (H) Mostly Marked Population, and (I) Nest Focused were suboptimal (dominated) solutions because alternative strategies provided greater benefits, lower cost, or both. This analysis suggests that if the most expensive strategy is not affordable, strategies A, B, and C are good options. If only the least expensive options are possible, strategy D may be the preferred option.

Weights applied to objectives and risk attitudes affected the optimal solution. In the scenario with an equal weight on the six fundamental objectives and a risk neutral decision maker (Figure 6a), strategy (G) Marked Population had the greatest utility. Strategy (G) provided the greatest utility for monitoring the recovery criteria for a risk neutral decision maker, but was the most expensive strategy among the alternatives (Figure 6a). For a risk averse decision maker, however, strategy (B) Varied Population Sizes had the greatest utility under the equal weights scenario (Figure 6b). Strategies (A) and (C) were also competitive when risk aversion was appropriate. The risk premium for the recovery criteria and cost thus resulted in strategies (A), (B), and (C) being as good or better than the most expensive strategy (G; Figure 6b). The results were qualitatively similar under the “more on mandates” weight scenario which emphasized the recovery criteria (population size and productivity) and cost (80% of the weight split evenly among the three). Strategy (G) Marked Population again had the greatest utility in a risk neutral setting, largely due to the expected high accuracy of monitoring the recovery criteria with this strategy (Figure 6c). For a risk averse decision maker and more weight on mandates, strategies A, B, and C all provide cost effective monitoring and greater utility than strategy G (Figure 6d).

### 3.4. Sensitivity of Monitoring Strategies to Objective Weights

The weight assigned to cost reflects how important cost is to a decision-maker when choosing a monitoring strategy to implement. We conducted a sensitivity analysis from the perspective of a risk averse decision maker (Figure 7). In general, and not surprisingly, as the weight assigned to cost increased, the utility of the most expensive strategy (G) decreased and the utility of the least expensive strategies increased. However, the strategy with the greatest total utility was robust to the weight assigned to cost over a fairly large range. If the weight assigned to cost was between approximately 0.15 and 0.45, the highest-ranking strategy was (B) Varied Population Size. If the decision maker preferred less weight for cost (i.e., <0.15), the highest-ranking strategy would be (G) Marked Population, which is the most expensive strategy among those we considered (Figure 5 and Figure 6). If the decision maker preferred more weight on cost (above approximately 0.45), the highest-ranking strategy would be (D) Minimal Marked Population, the second least expensive strategy among those we considered.

## 4. Discussion

The need is great for a reliable monitoring program to discern the status of an at-risk species, and for a standardized and feasible approach to monitoring populations of most any threatened or endangered species to inform down- or delisting decisions. Assessing how a species may be responding to changing environmental conditions, determining whether changing environmental conditions may be positively or negatively altering a species’ risk of extinction, and communicating those conclusions are all improved with the clarity that results from scientifically and statistically sound evidence. Without standardization, however, monitoring results might not be easily compared across a species’ range or over time and can make it more challenging to assess the best available information, and determine the degree to which a species status may be improving or declining. In the case of WSP populations, we expect that a comprehensive assessment and means of evaluating options for monitoring across the multiple Recovery Units and associated agencies and institutions engaged in conservation efforts may be able to promote the increased consistency in the methods and degrees of monitoring intensity, and improved clarity around the resulting outcomes.

We provided this analysis as a basis for USFWS to evaluate the efficacy and relative costs of alternative WSP monitoring strategies for consideration for range-wide standardization and implementation. We have produced a set of evaluations along with simple analysis tools (computer spreadsheets) that can be used by managers or decision-makers to explore the implications of alternative risk attitudes defining various weights assigned to monitoring and recovery objectives, on utility values of nine alternative monitoring sampling strategies. We also emphasized the value of retaining and displaying the individual expert panelist scores, and the variations among their scores, for each monitoring objective and monitoring strategy combination (Figure 4), to best express the variation among the panelists’ field experience, local field conditions, institutions, and other factors. We expressly did not have the panelists reach singular consensuses on score values, nor to eliminate outlier score values which may have high and authentic values representing some unique panelist’s experience or field situation.

Monitoring WSP populations includes a number of additional field and office activities not specifically represented in the list of monitoring objectives and methods of each sampling strategy presented here, and likely could also be accounted for when selecting a strategy for standard application. For example, additional duties associated with WSP monitoring by the Oregon Biodiversity Information Center (ORBIC) and other institutions include [3]:Maintaining individual field journals during the breeding season.After the field season, compiling field notes into “nest cards” (individual nest histories) and a banding database, should banding of birds be conducted.Implementing quality control of denoting nest GPS locations, by reviewing maps for the final seasonal report.Maintaining field equipment including main vehicles, trailers, off-road and all-terrain vehicles, optics, cameras, exclosure fencing, and other gear.Preparing an annual field season monitoring report for presentation at the wide-range recovery meeting.Coordinating with other institutions and agencies, such as Wildlife Services, on a regular basis to report predator observations and predation events.In addition, coordinating with funding partners, including answering questions, attending meetings, and alerting land management agencies of unusual predator or violation activities.

As well, a number of specific field tasks and activities are associated with most monitoring strategies, such as erecting symbolic fencing around WSP nesting areas, deployment and maintenance of nest cameras if used, erecting and maintaining nest predator exclosures, salvage or orphaned chicks and eggs, etc. The various institutions and agencies involved in WSP monitoring may perform in different ways some monitoring activities in concert with others so as to save time and cost for field travel, and may integrate WSP population monitoring activities with other related tasks such as sampling and identifying invasive plants, and direct management of nest predators. Thus, although it was attempted, it became too difficult to develop absolute cost estimates of each and every activity under each sampling strategy explored, across all WSP recovery zones and institutions. Nevertheless, such costs are real and would need to be considered when selecting a sampling strategy to institute as a standard across all recovery zones.

We also recognize the value of working in cooperation with others to advance species recovery, and that the USFWS is compelled to engage with partners throughout the entire recovery process, from planning through implementation [34]. Additional communication, coordination, and collaboration with a wide variety of recovery partners could be beneficial to achieving broad adoption and consistent implementation of a standardized monitoring strategy. Communication relative to the evaluation and analysis tools described herein has already taken the form of presentations at annual meetings gathering WSP recovery partners from across the range of the species [35,36] and annual national wildlife conferences gathering wildlife professionals from across the country [37].

Our evaluation and analysis tools may prove most valuable to USFWS-led recovery efforts for choosing and explicitly incorporating a standardized monitoring strategy that is feasibly applied across the species range, and that could be expanded to include other potential monitoring objectives and approaches such as integrated multispecies monitoring [38]. As suggested previously in this article, recovery plans are central to identifying, coordinating, and prioritizing recovery actions, such as monitoring programs, and are important tools that ensure that sound scientific and strategic decision making occurs throughout a species’ recovery process. Keeping the WSP recovery plan current with an updated monitoring strategy can help ensure that the recovery plan continues to serve its purpose of providing the USFWS and its recovery partners with a roadmap to species’ recovery. More specifically, our decision-analytic approach can serve as a useful tool to communicate the need for certain sampling strategies for achieving species recovery objectives, and for re-evaluating conditions over time, and as a model for use with other species recovery projects requiring reevaluation of inventory, monitoring, and research activities. The monitoring strategies considered in this assessment varied in the degree to which they would provide accurate, timely, and actionable data on the population status.

## 5. Conclusions

The analysis entailed a structured decision-making process of evaluating the degree to which alternative strategies for sampling populations of WSPs could meet population monitoring objectives and recovery goals for the species. Our analysis provides the decision-maker with much background information on the expected performance of each strategy, and the capacity to determine the influence of emphasizing different performance measures and costs. We offer our analysis as a broad decision-science based analysis framework that can be adapted to new and changing conditions. We suggest flexibility in implementing any final decision from this work to account for potential, unforeseen changes in environments, populations, costs, objectives, etc.

Results of our work have been communicated to WSP program directors in USFWS via formal presentations of results and informal discussions for their decision-making consideration and implementation to ensure that monitoring methods are instituted consistently across the species’ range. Our use of structured decision-making could be applied to cases of other species once imperiled but now on the road to recovery.

## Figures and Tables

**Figure 1 animals-11-00569-f001:**
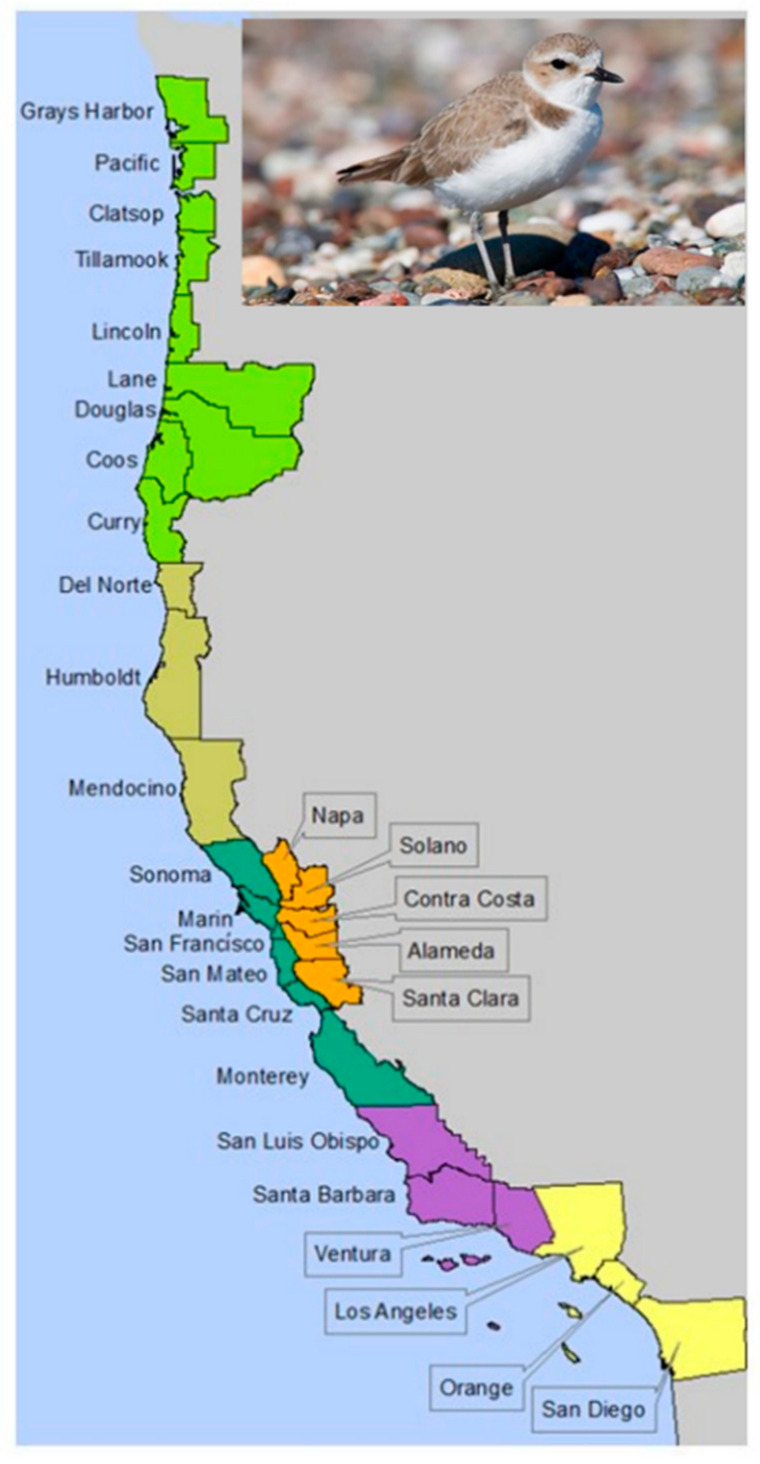
Western snowy plover recovery units (colors) and counties (polygons) along the west coast, US. The six recovery units include: (1) Washington and Oregon (lime green); (2) northern California (Del Norte, Humboldt, and Mendocino counties; tan); (3) San Francisco Bay (locations in Napa, Alameda, Santa Clara, and San Mateo counties; orange); (4) Monterey Bay (including coastal areas along Monterey, Santa Cruz, San Mateo, San Francisco, Marin and Sonoma counties; dark green); (5) San Luis Obispo, Santa Barbara, and Ventura counties (purple); and (6) Los Angeles, Orange, and San Diego counties (yellow) (photo source: Public domain, US Fish and Wildlife Service, Oregon Field Office, https://www.fws.gov/oregonfwo/articles.cfm?id=149489510 (accessed on 21 February 2021)).

**Figure 2 animals-11-00569-f002:**
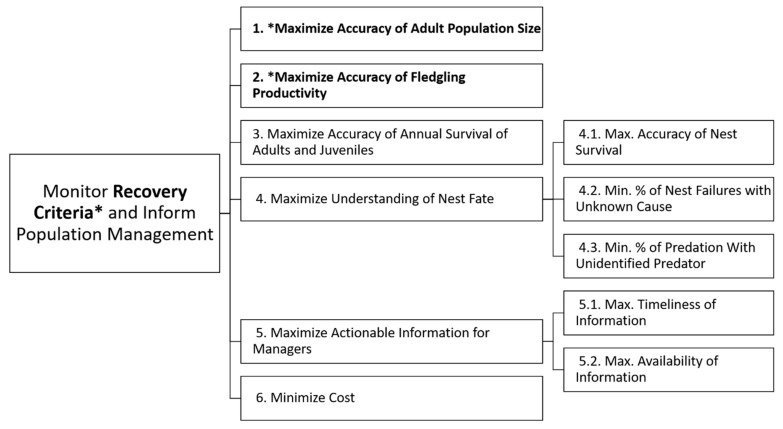
Hierarchy of objectives for monitoring Western snowy plovers. The hierarchy was created by a technical review team representing decision makers and stakeholders. The hierarchy has six fundamental objectives and two of the objectives have sub-objectives. See Table 1 for the weight scenarios which were investigated. Recovery criteria are indicated with an asterisk *.

**Figure 3 animals-11-00569-f003:**
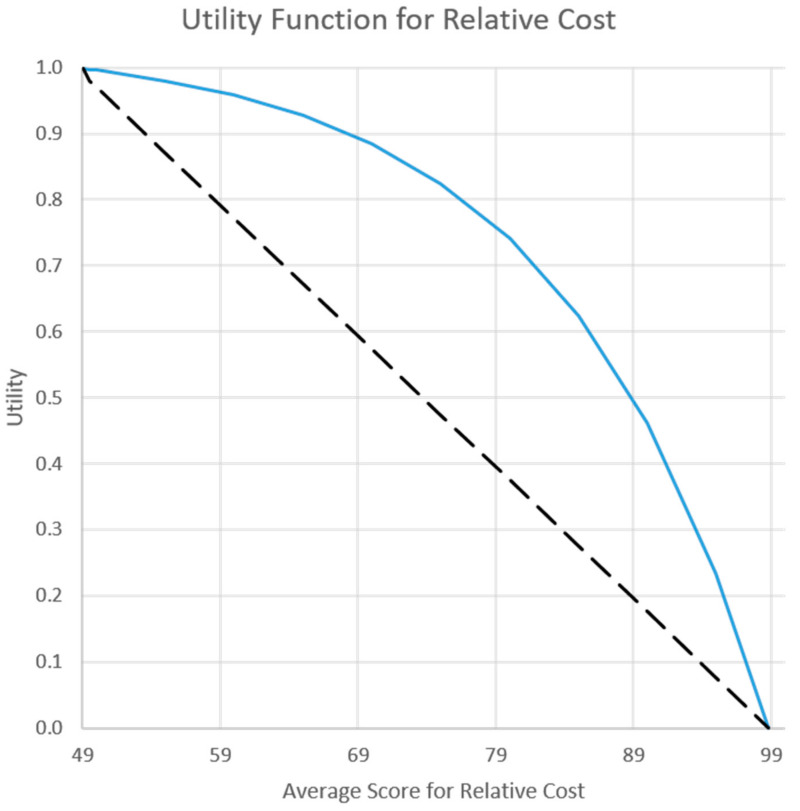
Utility functions for cost showing risk neutral (dashed line) and risk averse (solid line) decision makers.

**Figure 4 animals-11-00569-f004:**
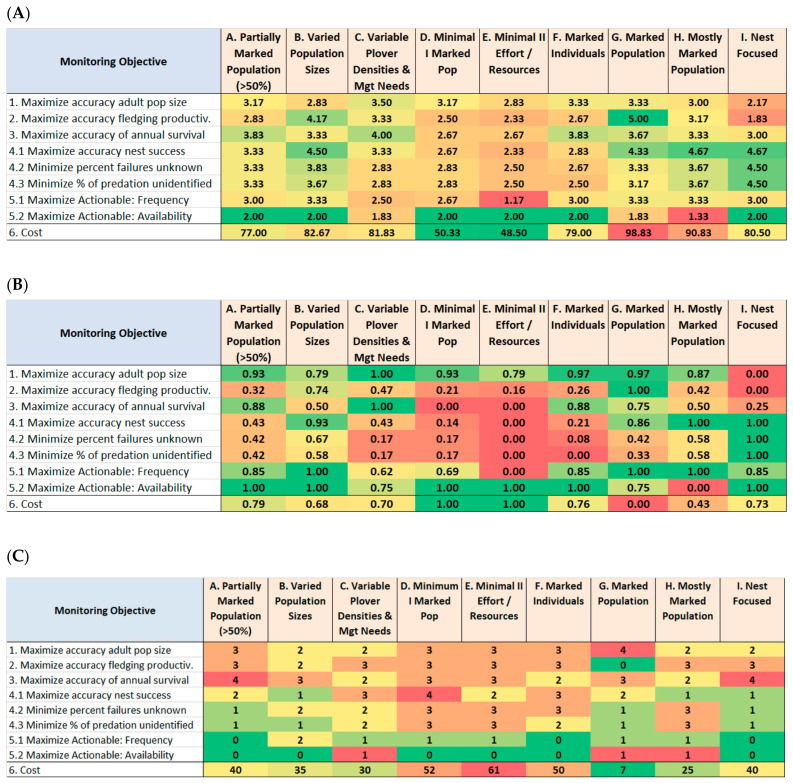
Summary of mean values of the utility scores of the sampling strategies, by monitoring objective, as averaged across the six members of the WSP Technical Team. (**A**) Means of raw scores; (**B**) scores normalized across the sampling strategies for each monitoring objective; (**C**) ranges of raw scores. Green indicates best outcomes (highest utilities or lowest cost or smallest range), red indicates worst outcomes (lowest utilities or highest cost or greatest range).

**Figure 5 animals-11-00569-f005:**
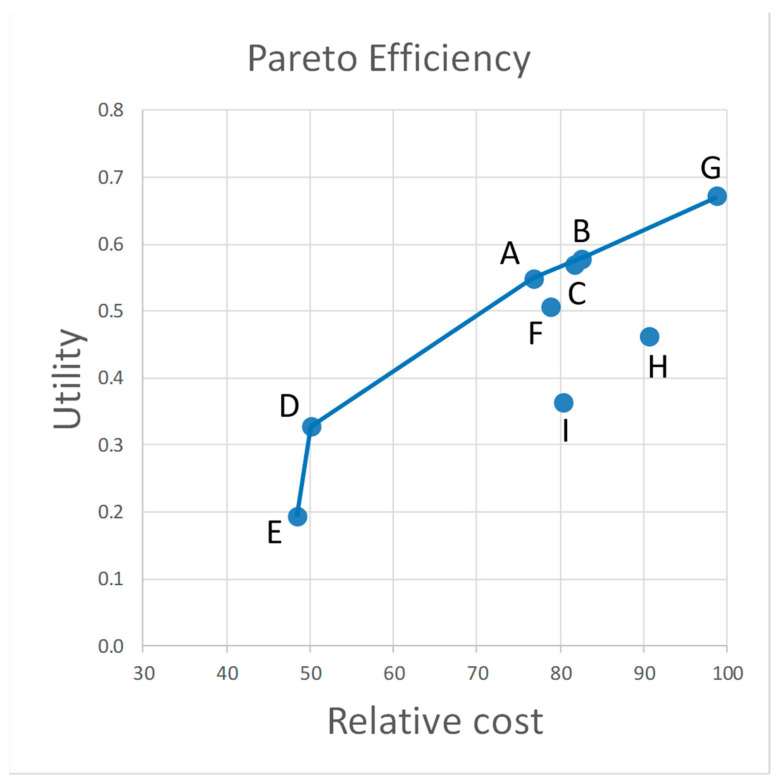
Pareto efficiency of competing monitoring strategies to evaluate Western snowy plover populations. Each symbol (A–I) is a monitoring strategy (Table 2). The solid line is the Pareto efficiency frontier showing the strategies with the greatest utility for a given level of cost, i.e., the non-dominated solutions. Monitoring strategies below the efficiency frontier (F, H, and I) are suboptimal choices because other strategies provide greater or equal utility for the same or lower cost. This analysis uses equal weight for six fundamental objectives and risk neutral decision makers (see Table 1 for objective weights).

**Figure 6 animals-11-00569-f006:**
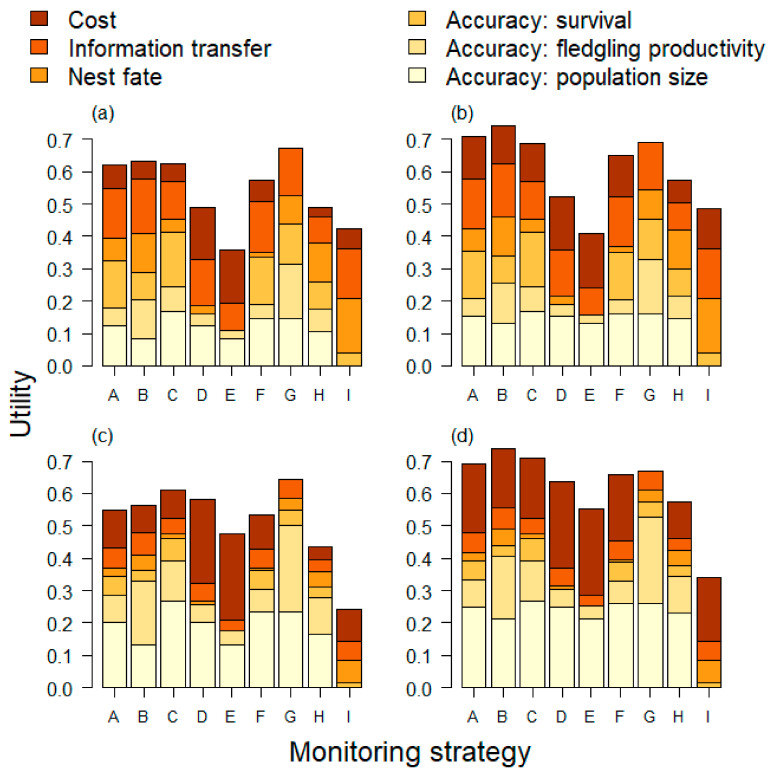
Summary of multi-attribute utility analysis. (**a**) The “equal” weight scenario (Table 1) with a risk neutral decision maker. (**b**) The “equal” weight scenario with a risk averse decision maker. (**c**) “More on mandates” weight scenario (Table 1) with a risk neutral decision maker. (**d**) “More on mandates” with a risk averse decision maker. In general, risk management shown in (**b**) and (**d**) changes the preferred monitoring strategy from G, the most expensive strategy, to B; strategies A and C are also competitive when risk management is added. The accuracy of population size and fledgling productivity (two lightest shades) are important for monitoring two recovery criteria for the species. See Table 2 for descriptions of the strategies and Appendix A for details.

**Figure 7 animals-11-00569-f007:**
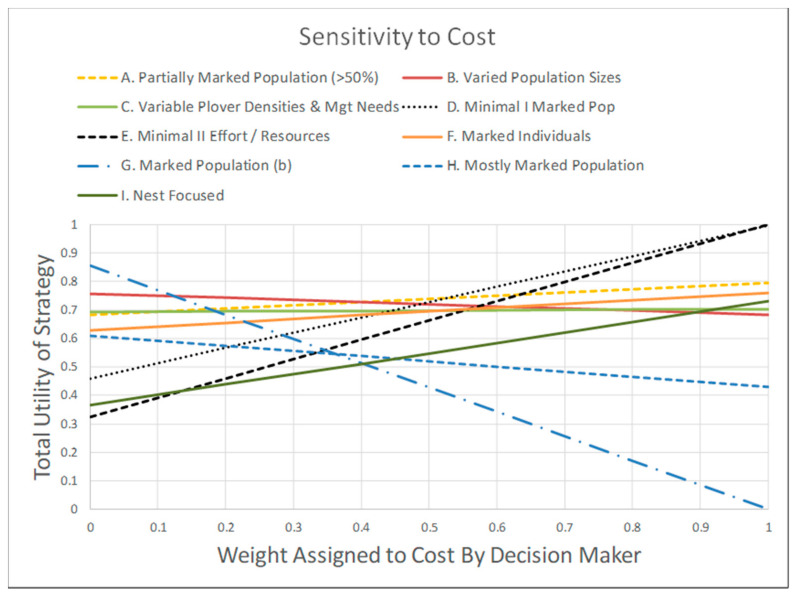
Sensitivity of total utility to the weight assigned to the cost objective. This analysis is for a moderately risk averse decision maker (as specified in Appendix A). If cost receives a weight between approximately 0.15 and 0.45 (x-axis), the preferred monitoring strategy is “B. Varied Population Sizes”. Outside this range, the preferred monitoring strategy was either “G. Marked Population (b)”, when the weight assigned to cost is < 0.15, or “D. Minimal I Marked Population”, when the weight assigned to cost is >0.45. The choice of strategy B was thus robust over a relatively large range in the weight assigned to the cost objective.

**Table 1 animals-11-00569-t001:** Monitoring objectives, performance measures, and weights used to evaluate potential monitoring strategies. Monitoring objectives 1 and 2 are related to the recovery criteria * and are considered monitoring mandates; for the purposes of this analysis, cost was also considered a mandate. The “Equal” objective weight scenario places equal weight on the six fundamental monitoring objectives; “more on mandates” places 80% on recovery mandates (objectives 1, 2, and 6) and 20% on the remainder (objectives 3–5). Our performance measures are constructed scales to quantify the achievement of objectives; see Appendix A for details of the constructed scales.

		Objective Weight Scenario	
Monitoring Objective	Performance Measure	“Equal”	“More on Mandates”
1. * Maximize accuracy adult population size	Accuracy: Bias and precision	0.167	0.267
2. * Maximize accuracy fledging productivity	Accuracy: Bias and precision	0.167	0.267
3. Maximize accuracy of annual survival	Accuracy: Bias and precision	0.167	0.067
4. Maximize understanding of nest fate			
4.1 Maximize accuracy of nest success	Accuracy: Bias and precision	0.056	0.022
4.2 Minimize percent failures unknown	Effectiveness: Identifying causes of nest failures	0.056	0.022
4.3 Minimize % of predation unidentified	Effectiveness: Apportioning sources of nest predation	0.056	0.022
5. Maximize information transfer to managers			
5.1 Maximize Actionable Info.: Timeliness	Frequency of reports to managers	0.083	0.033
5.2 Maximize Actionable Info.: Availability	Type of report to managers	0.083	0.033
6. Minimize cost	Relative cost	0.167	0.267
Total		1.0	1.0

**Table 2 animals-11-00569-t002:** Strategy table used to summarize nine potential monitoring strategies (A to I) and methods identified by the Western snowy plover technical team for analysis (also see Appendix A for detailed descriptions of each monitoring strategy).

Monitoring Strategy	Population Size	Fledgling Productivity ^a^	Annual Survival	Nest Fate	Communicate Information to Managers
A. Partially Marked Population (> 50%)	Multiple breeding window surveys at set time intervals. N-mixture models used to correct counts.	Band sample of chicks at each site.	Repeated counts of unmarked individuals with some mark-recapture.	Monitor a sample of nests across time and space, physically examine every 3 days. Use nest cameras to identify predators.	Weekly conference call.
B. Varied Population Sizes	Multiple breeding window surveys. Banded breeder adjusted with nest ownership.	At large sites, band a random sample of chicks; at small sites, band all chicks.	Periodic mark-resight at banding sites.	Monitor sample of nests at sites where λ < 1 ^b^ Use nest cameras to identify predators.	Ad-hoc conference calls with managers.
C. Variable Plover Densities and Management Needs	Multiple breeding window surveys.	Band a sample of chicks; weekdays only to reduce disturbance.	Repeated unmarked counts with mark-recapture.	Monitor every nest sample at sites ^c^ with ≤4 pairs, physically check some at least every 3 days, others > every 3 days. Use nest cameras to identify predators.	Weekly and ad-hoc conference calls.
D. Minimal I Marked Population	Multiple breeding window surveys.	Band a subset of chicks on sites where the regional λ < 1.	Repeated counts of unmarked individuals. Model survivorship with open population N-mixture models.	Monitor sample based on environmental factors and/or λ < 1, at least every 3 days. Use nest cameras to identify predators.	Weekly report at sites where λ < 1.
E. Minimal II Effort/Resources	Multiple breeding window surveys.	Band subset of chicks where λ < 1.	Do not monitor.	Sample of nests at least every 3 days.	Monthly report.
F. Marked Individuals	Banded breeder adjusted with counts.	Band a sample of chicks (all individuals at a subset of dedicated sites).	Mark-resight, banding all individuals at a subset of dedicated banding sites.	Monitor a sample of all nests across time and space.	Combination of individual and daily, weekly, monthly, and ad-hoc conference calls; also preparation and distribution of weekly, monthly, annual, and ad-hoc monitoring reports.
G. Marked Population	Peak count of nests and broods. Banded breeder adjusted with counts.	Attempt to band all chicks within the study area.	Mark-resight, banding all individuals.	Monitor every nest at least every 3 days. No cameras.	Combination of daily, weekly, monthly, and ad-hoc conference call; also annual and weekly reports.
H. Mostly Marked Population	Banded breeder adjusted with counts.	Band sample of chicks.	Mark-resight, banding a subset determined through power analysis.	Monitor every nest at intervals > every 3 days. Use nest cameras to identify predators.	Daily communication with individuals.
I. Nest Focused	Peak count of nests and broods.	Calculate average fledging rate based on number of chicks hatched.	Repeated occupancy surveys. Model survivorship using dynamic N-occupancy models.	Monitor every nest at least every 3 days. Use nest cameras to identify predators.	Monthly conference call with weekly and annual reports.

^a^ Fledgling productivity is the number of fledged young per adult male. ^b^ λ: Population annual growth rate. ^c^ Sites are defined in Appendix L of the Recovery Plan [2].

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
