# Peer review of "Using Decision Science for Monitoring Threatened Western Snowy Plovers to Inform Recovery"

_animals, 2021, doi:10.3390/ani11020569_

Round 1

Reviewer 1 Report

This is a well written, well thought-out paper that presents a structured decision making approach to monitor the WSP population, given that it is increasing.  I think the paper is sound, the writing style is strong and the figures are excellent; it should be published.  But I also think the authors should consider and reflect upon a few more issues and incorporate them into the Discussion and/or Conclusions.  For example, how may the different options actually work in the field, given logistics, degrees of expertise, who actually performs field work (e.g. interns? volunteers? professional biologists?).  In addition, some reflection on the potential sensitivity of different approaches on any population trend analyses detected over time is in order.  Also, what about considering options in which the USFWS only surveys populations every other or every third year as it grows, given that the population is threatened (not endangered) and recovering, rather than annually?  Would this likely change any recommendations as to which approach is optimal or economically more viable?  I would also like to see a bit more about individual variation in expert opinion, which may allow the authors to consider and recommend bout how many experts would be needed (at a minimum) to reach some consensus.  While none of these were direct objectives of the study, and the paper represents a planning exercise as opposed to a field test of it, I think that some more reflection on real-world issues would enhance the discussion and increase its relevance to readers and managers.

Author Response

  1. Comments and Suggestions for Authors - From Animals Reviewer No. 1

This is a well written, well thought-out paper that presents a structured decision making approach to monitor the WSP population, given that it is increasing.  I think the paper is sound, the writing style is strong and the figures are excellent; it should be published.  

Authors' response:  No edit or response needed.

But I also think the authors should consider and reflect upon a few more issues and incorporate them into the Discussion and/or Conclusions.  

Authors' response:  We address each of these reviewer comments individually as follows.

For example, how may the different options actually work in the field, given logistics, degrees of expertise, who actually performs field work (e.g. interns? volunteers? professional biologists?).  

Authors' response:  We chose not to greatly expand the manuscript by trying to describe potential field operations of individuals, which was not the clearly stated purpose of the paper.  To address this with any usefulness would entail a very detailed analysis of the administrative structures and working relations of each agency and institution.  It is not the purpose of this paper to suggest specific operational protocols for any such institution or agency.  We do, however, already address that results of the analysis have been passed on to, and presented to, those bodies, as we state "Communication relative to the evaluation and analysis tools described herein has already taken the form of presentations at annual meetings gathering WSP recovery partners from across the range of the species (Todd et al., 2016; Todd 2017) and annual national wildlife conferences gathering wildlife professionals from across the country (Marcot et al., 2019)." 

In addition, some reflection on the potential sensitivity of different approaches on any population trend analyses detected over time is in order.  

Authors' response:  This is a good point.  However, the panelists' score values already essentially speak to this point, that is, the degree to which each monitoring objective -- gauged through their corresponding performance measure -- would far under each monitoring strategy.  That's sort of the whole point of their scoring:  the scores essentially speak to ability to detect trend.  Table 1 clearly denotes how each monitoring objective was evaluated in terms of performance measures that relate explicitly to accuracy, bias, precision, effectiveness, and other criteria.  We discussed the strategies in terms of an annual implementation, but trend is based on annual estimates, and the accuracy (bias and precision) of the strategy will determine power to detect trends.  Further,

the sensitivity to detect trend was the underlying motivation for using accuracy as an objective, and the way we defined the performance measure as a combination of bias and precision already covers this.

Also, what about considering options in which the USFWS only surveys populations every other or every third year as it grows, given that the population is threatened (not endangered) and recovering, rather than annually?  Would this likely change any recommendations as to which approach is optimal or economically more viable?  

Authors' response:  We chose not to greatly expand the number of monitoring strategies beyond those evaluated, for the reason that they were devised in collaboration with the Western Snowy Plover Technical Team that consisted of plover monitoring experts.  The paper provides an overall process that could be adapted for evaluating other monitoring strategies should the relevant agencies and institutions so choose. 

I would also like to see a bit more about individual variation in expert opinion, which may allow the authors to consider and recommend bout how many experts would be needed (at a minimum) to reach some consensus.  

Authors' response:  In Results section 3.2, we already address this to some degree, concluding with "Thereafter, any further apparent disparities in scoring among the team members were appropriate and authentic outcomes of members differing in their field experience, in how their institutions conduct monitoring, and other practical matters.  To this end, we retained all such differences in scoring in our analyses, and we did not call for consensus or eliminate outlier score values."  That is, we expressly did not seek consensus, or to "eliminate outlier score values," as suggested by this reviewer.  To this end, we added some text to Discussion:  "We also emphasized the value of retaining and displaying the individual expert panelist scores, and the variations among their scores, for each monitoring objective and monitoring strategy combination (especially Fig. 4c, that displays ranges of the raw scores among the panelists), to best express the variation among the panelists' field experience, local field conditions, institutions, and other factors.  We expressly did not have the panelists reach singular consensuses on score values, nor did we seek to eliminate outlier score values which may have high and authentic value representing some unique panelist's experience or field situation."

While none of these were direct objectives of the study, and the paper represents a planning exercise as opposed to a field test of it, I think that some more reflection on real-world issues would enhance the discussion and increase its relevance to readers and managers.
Authors' response:  We do already address a fairly long list of "real-world issues" in Discussion (see the 7 bullet-point list there).  We are unclear what additional "real-world issues" this reviewer comment is alluding to, but we think we have this issue already covered.  We feel that any "field test" would warrant a further exploration and a different manuscript. 

Reviewer 2 Report

This is an excellent manuscript on the use of decision analysis for deciding on appropriate monitoring approaches for a species in recovery. There is not much that I can comment on really. A couple of minor comments:

Figure 1. For an international audience it would be nice to have an insert map of where this geographic region sits in relation to the rest of North America 

Line 107: The example here seems a bit odd. Maybe either adding more details about this example or just referring to a few different examples (just using citations after citing Runge 2011).

Line 138: I assume "FWS" is the same as "USFWS" - if so then perhaps keep the abbreviation the same to avoid confusing the international audience

Line 194: One thing that is not clear to me is do the experts have experience of the different monitoring strategies and how do they, for example, consider accuracy - are they thinking in terms of comparing "model" estimates to observations?

Author Response

  1. Comments and Suggestions for Authors - From Animals Reviewer No. 2

This is an excellent manuscript on the use of decision analysis for deciding on appropriate monitoring approaches for a species in recovery. There is not much that I can comment on really.

A couple of minor comments:

Figure 1. For an international audience it would be nice to have an insert map of where this geographic region sits in relation to the rest of North America 
Authors' response:  We have not included an inset map of North America, but we do clearly explain the geographic region of interest in the text and in the figure 1 caption, so all the locational information is provided.  However, if the journal editor requests such a map, we can provide it. 

Line 107: The example here seems a bit odd. Maybe either adding more details about this example or just referring to a few different examples (just using citations after citing Runge 2011).
Authors' response:  We made this edit. 

Line 138: I assume "FWS" is the same as "USFWS" - if so then perhaps keep the abbreviation the same to avoid confusing the international audience
Authors' response:  We made this edit. 

Line 194: One thing that is not clear to me is do the experts have experience of the different monitoring strategies and how do they, for example, consider accuracy - are they thinking in terms of comparing "model" estimates to observations?

Authors' response:  Good point; we clarified this in Methods to note that the experts do indeed have experience in different monitoring strategies.  And we clarify the interpretation of the use of
"accuracy" in the performance measures of the monitoring objectives (Table 1). 

Reviewer 3 Report

Thank you for the opportunity to comment here. I confess I am new to this process, and the analysis therein, it is complex and staged, and I hope a reviewer with specific experience of the analysis can also provide feedback.

I have provided minor comments on the ms, and would like to float a few other considerations:

  • A flowchart of the basic process would be helpful, even if it is in Supplementary Material. Adding Rho and lamda to the glossary would be helpful.
  • I wonder if what is discussed is the minimum required monitoring, with “above baseline” monitoring possible where cheap, free or useful? Would additional data be incorporated, or discarded?
  • In terms of objectives, a few seem missing (I understand these come from the recovery objectives). Firstly, some level of consistency with previous monitoring can be especially helpful, remembering that sometimes population trends are of interest, vary between regions, and that monitoring often is called upon to fulfil additional objectives which were never anticipated – I understand that is not desirable but it happens all the time. So, for example, the census of WSPs presumably provides lots of data on species other than WSP, and in future that data might well be critical. What if the recovery objectives change in future? Perhaps the idea of versatility could also be at least mentioned? At least some consistency would be critical I would have thought. Secondly, some information about timeframes for objectives could be useful, i.e. the temporal scale at which objectives operate (some timescales on communication is available).
  • While a transparent process to provide informed choices has been presented, the outcome is not validated, and the choice could be bad either because the inputs are not reliable or because we do not understand how populations work. This needs to be mentioned.
  • Things can change, and that may alter which choice was optimal, or would be optimal into the future, and may necessitate going to a different option which would then need to be “backward compatible”. Possible changes include:
    • Emerging threats could take us by surprise (Hurricane Sandy and Piping Plovers? Maslo, Brooke, Karen Leu, Todd Pover, Michael A. Weston, Ben L. Gilby, and Thomas A. Schlacher. "Optimizing conservation benefits for threatened beach fauna following severe natural disturbances." Science of the total environment649 (2019): 661-671.) and change how we wish to monitor.
    • Costs can change, some may go up, others may go down. This could involve changes to staff training, or even new statistical modelling techniques.
    • Capacity may change, what is simple now may be rendered impractical in the future.
    • The social licence may change. This could involve changes to ethical aspects, first nations changes (in Australia, recent concerns over Traditional Owner Sacred Sites, and uncertainty about the extent of those sites, has prevented some pairs of Hooded Plovers being managed as they were previously).
    • The distribution may change (see the ref on Sandy) or because of sea level rise etc.
  • It is therefore surprising that there is not a firm recommendation to either “trial” any option to test it, or undertake some form of validation. Surely some kind of periodic review is needed against the objectives of the monitoring? As a scientist, I wonder if there is already data to compare the performance of the different models, including quantifying their actual costs? If not, it could be run as an experiment, at least initially?
  • In a few places the authors note aspects that were “beyond the scope” of the current study, which is fine, but some of the alternatives, such as integrated multispecies monitoring (Hansen, B. D., Clemens, R. S., Gallo‐Cajiao, E., Jackson, M. V., Kingsford, R. T., Maguire, G. S., Weller, D. R. Golo Maurer, David Milton, Danny I. Rogers, Dan R. Weller, Michael A. Weston , Eric J. Woehler & Richard A. Fuller (2018). Shorebird monitoring in Australia: A successful long‐term collaboration among citizen scientists, governments and researchers. Monitoring threatened species and ecological communities, 149-164.), might be useful to at least have flagged with readers.

I hope this helps, and sorry I am not more use on the analysis side of things. Mike Weston

Author Response

  1. Comments and Suggestions for Authors - From Animals Reviewer No. 3

Thank you for the opportunity to comment here. I confess I am new to this process, and the analysis therein, it is complex and staged, and I hope a reviewer with specific experience of the analysis can also provide feedback.

I have provided minor comments on the ms, and would like to float a few other considerations:

A flowchart of the basic process would be helpful, even if it is in Supplementary Material.
Authors' response:  We suggest that we adequately explained the procedures sequentially in Methods, and that a single flowchart could become very complex to denote all steps in the process.  We would defer to the journal editor and could attempt this if the editor felt it imperative. 

Adding Rho and lamda to the glossary would be helpful.
Authors' response:  We removed the supplemental appendix glossary as we do not cite it in the text.

I wonder if what is discussed is the minimum required monitoring, with “above baseline” monitoring possible where cheap, free or useful? Would additional data be incorporated, or discarded?
Authors' response:  Our approach is designed to deliver the monitoring data that is required by the Recovery Plan in the most cost-effective manner. The monitoring strategies we evaluated in the analysis would not be considered "minimum" from this perspective because we included monitoring activities (related to annual survival, nest fate, and information for managers) that are more than what is mandated in the Recovery Plan. We have not suggested that additional data beyond what is included in our stated monitoring objectives would be discarded. Additional data collection beyond what is identified in our monitoring objectives, and data management for any such data, were not explicitly treated in our analysis. Resources necessary for data management related to our specified monitoring objectives were implicit in the cost estimates we used, but the details of how additional data beyond what is mandated by the Recovery Plan would be handled is beyond the scope of this study.

In terms of objectives, a few seem missing (I understand these come from the recovery objectives). Firstly, some level of consistency with previous monitoring can be especially helpful, remembering that sometimes population trends are of interest, vary between regions, and that monitoring often is called upon to fulfil additional objectives which were never anticipated – I understand that is not desirable but it happens all the time. So, for example, the census of WSPs presumably provides lots of data on species other than WSP, and in future that data might well be critical. What if the recovery objectives change in future? Perhaps the idea of versatility could also be at least mentioned? At least some consistency would be critical I would have thought. Secondly, some information about timeframes for objectives could be useful, i.e. the temporal scale at which objectives operate (some timescales on communication is available).
Authors' response:  We adhered, as stated, to the objectives explicitly stated in the species' recovery plan, and did not embellish or add to those objectives on our own.  We also did not speculate about change in any such recovery objectives; at least as of this writing, there are no stated intensions or plans by USFWS to amend those recovery objectives for this species.  We provide an overall decision-science based assessment framework that could be applied to other recovery objectives, as needed, and we state this "versatility" in the paper.  Also, we are unclear what the reviewer here means by "At least some consistency would be critical I would have thought"; consistency of what, with what?  As for timescales of objectives, this is not part of the analysis, which is not at all intended to be a critique of the existing recovery objectives, but rather an analysis of the efficacy by which alternative strategies could meet those existing published objectives. 

While a transparent process to provide informed choices has been presented, the outcome is not validated, and the choice could be bad either because the inputs are not reliable or because we do not understand how populations work. This needs to be mentioned.
Authors' response:  We are confused by this comment.  The manuscript is not an analysis and critique of population monitoring methods nor or the biologists involved in this work.  We state that we chose the team members based on their empirical field experience in various monitoring methods and analyses.  Further, we do not understand the comment on validation; if the reviewer is asking for our decision-analysis method itself to be "validated," we point to the introduction where we cite a number of references on the use of decision science approaches to species recovery and on studies of this species per se.  We also do not quite follow the point that "we do not understand how populations work."  The suite of monitoring strategies we analyzed are all well-founded and traditionally applied in many other population monitoring situations. 

Things can change, and that may alter which choice was optimal, or would be optimal into the future, and may necessitate going to a different option which would then need to be “backward compatible”. Possible changes include:

  • Emerging threats could take us by surprise (Hurricane Sandy and Piping Plovers? Maslo, Brooke, Karen Leu, Todd Pover, Michael A. Weston, Ben L. Gilby, and Thomas A. Schlacher. "Optimizing conservation benefits for threatened beach fauna following severe natural disturbances." Science of the total environment649 (2019): 661-671.) and change how we wish to monitor.
  • Costs can change, some may go up, others may go down. This could involve changes to staff training, or even new statistical modelling techniques.
  • Capacity may change, what is simple now may be rendered impractical in the future.
  • The social licence may change. This could involve changes to ethical aspects, first nations changes (in Australia, recent concerns over Traditional Owner Sacred Sites, and uncertainty about the extent of those sites, has prevented some pairs of Hooded Plovers being managed as they were previously).
  • The distribution may change (see the ref on Sandy) or because of sea level rise etc.
  • It is therefore surprising that there is not a firm recommendation to either “trial” any option to test it, or undertake some form of validation. Surely some kind of periodic review is needed against the objectives of the monitoring? As a scientist, I wonder if there is already data to compare the performance of the different models, including quantifying their actual costs? If not, it could be run as an experiment, at least initially?

Authors' response:  Again, our work provides an overall decision-science based assessment framework that could be applied to other recovery objectives, as needed, and we state this in the paper.  We agree that there can be unforeseen changes in the environment and population conditions, that costs might change, and sea levels might rise, and the other factors mentioned above might ensue.  We do add text to the Conclusion section now noting the good point about implementing any final decision from this work with flexibility to account for unforeseen changes in environments or populations, and that re-evaluations of the decision under those changed conditions can be undertaken. 

In a few places the authors note aspects that were “beyond the scope” of the current study, which is fine, but some of the alternatives, such as integrated multispecies monitoring (Hansen, B. D., Clemens, R. S., Gallo‐Cajiao, E., Jackson, M. V., Kingsford, R. T., Maguire, G. S., Weller, D. R. Golo Maurer, David Milton, Danny I. Rogers, Dan R. Weller, Michael A. Weston , Eric J. Woehler & Richard A. Fuller (2018). Shorebird monitoring in Australia: A successful long‐term collaboration among citizen scientists, governments and researchers. Monitoring threatened species and ecological communities, 149-164.), might be useful to at least have flagged with readers.
Authors' response:  Very good, we added text and this citation accordingly. 

Specific comments and questions were embedded in the PDF ms for Table 2.

Authors' response:  We have addressed every comment and question therein, and amended some of the table text, and added several explanatory footnotes as requested. 

        One reviewer comment, however, was in row H. Mostly Marked Population, in the column Fledgling Productivity.  Our table entry there reads "Band sample of chicks."  The reviewer comment here is "I wonder if power may change over time."  Our response to this specific comment is:  This conjecture about a change in power over time implies a change in variance of annual survival over time. For the purposes of this study, we assume a more standard power analysis in which variance estimates are available and considered stationary. Exploration of the implications of non-stationary variance in annual survival is beyond the scope of this study.  We did not change the table text entry here.

Reviewer #3 also made a number of comments and edits within the PDF of the manuscript.

Authors' response:  We addressed every comment by following the suggested edits, or have otherwise addressed the issues within our annotated authors' responses herein.